# Silicone engineered anisotropic lithography for ultrahigh-density OLEDs

Hyukmin Kweon[1,9], Keun-Yeong Choi[2,9], Han Wool Park[1,9], Ryungyu Lee[2], Ukjin Jeong [1], Min Jung Kim[3], Hyunmin Hong[3], Borina Ha[1], Sein Lee [4], Jang-Yeon Kwon [4], Kwun-Bum Chung[3], Moon Sung Kang [5,6], Hojin Lee[2,7] ✉ & Do Hwan Kim [1,8] ✉

Ultrahigh-resolution patterning with high-throughput and high-fidelity is highly in demand for expanding the potential of organic light-emitting diodes (OLEDs) from mobile and TV displays into near-to-eye microdisplays. However, current patterning techniques so far suffer from low resolution, consecutive pattern for RGB pixelation, low pattern fidelity, and throughput issue. Here, we present a silicone engineered anisotropic lithography of the organic light-emitting semiconductor (OLES) that in-situ forms a non-volatile etch-blocking layer during reactive ion etching. This unique feature not only slows the etch rate but also enhances the anisotropy of etch direction, leading to gain delicate control in forming ultrahigh-density multicolor OLES patterns (up to 4500 pixels per inch) through photolithography. This patterning strategy inspired by silicon etching chemistry is expected to provide new insights into ultrahigh-density OLED microdisplays.

Owing to their excellent color purity, thin form factor, near-instant response time, and wide color gamut, organic light-emitting diodes (OLEDs) have not only led to commercialized displays (e.g., televisions, mobile phones, and tablets), but have also facilitated the development of a new class of microdisplays essential for augmented, virtual, mutual, and extended reality (AR/VR/MR/XR) devices-based metaverse[1–5]. In particular, for the realization of the wide range of OLED technologies, a precise pixelation technique that can implement fine red/green/blue (RGB) pixels with high production and reliability regardless of the dimensional form factor of the display is strongly required. To address this, several methodologies have been developed for producing fine RGB pixels in the OLEDs without causing a deterioration of their intrinsic optoelectronic properties. Fine metal masks (FMMs) method is typically used in the pixelation of commercialized OLEDs. However, the scalability of the FMMs to large-area displays or high-resolution RGB patterning (limited to below 1000 pixels per inch

(ppi)) is constrained because of mask sagging, shadow effect, and cross-contamination between the RGB pixels[5–7]. Furthermore, the productivity of the FMMs could become more aggravated as the RGB pixel resolution increases[8,9].

Alternatively, inkjet printing[10–12], orthogonal photolithography[13,14], and template-directed growth[5,15] have been explored. However, these approaches continue to suffer from consecutive RGB pixelation, restricted resolution (not acceptable down to RGB pattern and pitch size of 5 μm), pixel reliability, or throughput issues. Also, to realize high-resolution OLEDs without the direct patterning process of the RGB emitting layers, a concept of metasurface mirror-induced color representation has been explored[5]; however, it is required of sophisticated patterning techniques such as nanoimprinting or e-beam lithography for each RGB meta-mirror that inherently has scalability limit to be used in commercial OLEDs, and suffers from not only angular color shift, but also sideband emission from the common

[1]Department of Chemical Engineering, Hanyang University, Seoul 04763, Republic of Korea. [2]School of Information Communication Convergence Technology, Soongsil University, Seoul 06978, Republic of Korea. [3]Division of Physics and Semiconductor Science, Dongguk University, Seoul 04620, Republic of Korea. [4]School of Integrated Technology, Yonsei University, Incheon 21983, Republic of Korea. [5]Department of Chemical and Biomolecular Engineering, Sogang University, Seoul 04107, Republic of Korea. [6]Institute of Emergent Materials, Sogang University, Seoul 04107, Republic of Korea. [7]School of Electronic Engineering, Soongsil University, Seoul 06978, Republic of Korea. [8]Institute of Nano Science and Technology, Hanyang University, Seoul 04763, Republic of Korea. [9]These authors contributed equally: Hyukmin Kweon, Keun-Yeong Choi, Han Wool Park. ✉e-mail: hojinl@ssu.ac.kr; dhkim76@hanyang.ac.kr

emissive layer, resulting in the degradation of color purity. This indicates that the development of an alternative patterning methodology of the OLES for ultrahigh-density RGB OLEDs, in conjunction with rational material design for high compatibility with the patterning process, is highly in demand.

Recently, direct photolithographic patterning of organic semiconductors (OSCs) has been demonstrated by exploiting ultraviolet (UV)-sensitive photo-cross-linkable additives (Supplementary Table 1). In this approach, the OSC films with photo-crosslinkers are processed as conventional photoresists, enabling selective modulation of film solubility in UV exposure regions. Consequently, the soluble regions (uncrosslinked) are eliminated via a solvent washing process. Although this methodology enables the OSC patterns of micron-scale and multi-pattern processing, the realized pattern sizes (currently, the reported minimum pitch size and maximum pixel density are 4 μm and 848 ppi[16], respectively) are insufficient for the implementation of high-density OLEDs. This is because of the collapse or swelling effect in the side region of the pattern developed via the isotropic wet-etching process or by light traveling underneath the dark area between the defined patterns of the photomask resulting from direct UV irradiation on the OSC layer[17]. Consequently, undesired trapezoid- or gaussian-shaped patterns were obtained, which makes it difficult to achieve the ultrafine pitch size of patterns[6,18–20]. This is a significant drawback for high-resolution displays requiring an ultrahigh pixel density (over 1000 ppi) because the pattern pitch size is the primary determinant of display resolution.

Alternatively, the application of conventional photolithography in combination with the reactive ion etching (RIE) dry process could be more beneficial in addressing well-defined high-resolution pixelation in the display industry. This is because the RIE-coupled photolithography (RCP) process currently adopted in the silicon industry effortlessly enables the generation of well-defined regular patterns, even down to sub-micron scales, with high-throughput productivity irrespective of dimension scales[21,22]. Notably, this exceptional patterning capability originates from the anisotropic etching chemistry of the RIE method at the desired regions to be etched. A key mechanism of the anisotropic etching in silicon thin films is the formation of a non-volatile etch-blocking layer (EBL), which is derived from the chemical reaction between silicon and reactive etching gases at the sidewall of the patterns[21,23,24]. The non-volatile EBL formed during the RIE process can effectively reduce the horizontal etch rate compared to the vertical etch rate, thereby enabling the implementation of highly elaborate patterns at high resolution. However, despite its effectiveness, this process has not been employed for organic light-emitting semiconductors (OLESs) in the OLEDs for both vacuum and solution deposition processes. This is because OLESs are intrinsically vulnerable to chemical and physical damage, which are inevitably encountered during the photolithography and RIE processes[7,25]. Thus, patterned shapes are distorted and the luminescence property of the OLESs rapidly deteriorates immediately after patterning. More importantly, the formation of non-volatile EBL is an exclusive chemistry of silicon-based materials, indicating that it is challenging to develop fine RGB pixels with anisotropic etching profiles in the OLESs.

Here, we report silicone-engineered anisotropic lithography of the OLES, which can facilitate the spontaneous formation of a non-volatile EBL into the film during the RIE process to realize ultrahigh-density OLEDs with well-defined RGB pixels. To achieve this, we designed solution-processable, silicone-incorporated OLES (SI-OLES) in which the OLES and silicone networks of bridged poly-silsesquioxanes (BPSQs) show three-dimensional molecular entanglement[26,27] (Supplementary Fig. 1). This becomes a cornerstone of physico-chemical tolerance of the SI-OLES for application of lithographic patterning process. In particular, silicone (Si-O-Si) blocks in the

BPSQs framework allow the SI-OLES to render a non-volatile EBL by inducing chemical reactions with reactive etching gases (Ar/O$_2$ or CF$_4$); thus, anisotropic lithography can be successfully implemented for ultrahigh pixelation of the OLEDs. On the basis of the unique features of the SI-OLES, we first demonstrate ultrahigh-density OLEDs corresponding to 4500 ppi. Moreover, even after the RCP process is applied, photoluminescence (PL) and electroluminescence (EL) characteristics of the SI-OLES are non-destructive, indicating that the proposed SI-OLES design can facilitate the practical realization of highly pixelated OLED microdisplays.

## Results and discussion
### Anisotropic etching characteristics of SI-OLES films
Silicon (Si) is a well-suitable material capable of developing anisotropic etching profiles by RIE-based dry etching. This is because a non-volatile EBL (e.g., Si$_x$O$_y$ or SiO$_x$F$_y$) is effectively formed on the sidewall of the etched trench via a chemical reaction between the silicon and gas radicals (e.g., O* or F*) (Fig. 1a). The sidewall-passivated chemistry of the silicon can attenuate the horizontal etch rate ($r_h$) by preventing lateral-directional chemical etching reactions. Meanwhile, the vertical etch rate ($r_v$) is accelerated via energetic ion bombardment, so that anisotropic etching of the silicon (f = 1 -($r_h/r_v$) ≈ 1) can be successfully introduced[22]. This non-volatile EBL-based anisotropic lithography, which has been regarded as a unique patterning behavior of the silicon, can be fully realized in the proposed SI-OLES (Fig. 1b). The silicone blocks of the SI-OLES enable the generation of the non-volatile EBL during the RIE process by reacting with gas radicals (Supplementary Note 1), resulting in well-defined vertical etch directionality. This groundbreaking method completely reverses the conventional concept that organic materials intrinsically exhibit isotropic etching behavior by generating volatile radical reactants in all etching directions. To provide the non-volatile EBL during the RIE process, the silicone molecules must be homogenously distributed in the SI-OLES films. The distribution of Si$^-$ signals corresponding to the silicone species was analyzed by time-of-flight secondary ion mass spectrometry (ToF-SIMS) with respect to sputtering time for the R- and G-SI-OLES films (Fig. 1c). The results show that the intensity of the Si$^-$ signals was identically obtained over the film thickness, indicating that the silicone networks were effectively incorporated in the OLES films without macro-phase separation. Notably, the film morphologies and PL/UV absorption properties of R- and G-SI-OLES films remained intact, despite the integration of the ladder-like silicone networks (Supplementary Figs. 2, 3). In addition, the SI-OLES films exhibited high chemical and dry etching resistance, indicating that a high-resolution photolithographic patterning process can be applied without deterioration of the pattern features and optoelectronic properties (Supplementary Figs. 4–6).

As a proof of concept, 2 μm × 6 μm stripe pixel micropattern arrays of the OLES and SI-OLES were prepared by utilizing the RCP process. The focused ion beam-scanning electron microscope (FIB-SEM) and fluorescence images of the micropatterns of the R-OLES/R-SI-OLES and G-OLES/G-SI-OLES were obtained (Fig. 1d and Supplementary Figs. 7–9). The micropatterns of the OLES showed significant side-etching abrasion, so that smaller and more undulated pattern features were obtained than the desired pattern dimension. By contrast, the SI-OLES films exhibited precise fine patterns with well-defined edge shapes, as determined by the pattern dimensions on the photomask. To investigate the precision of the SI-OLES-based micropatterns in detail, we quantitatively evaluated the key assessment factors of pattern fidelity, such as variation of pattern width and degree of pattern edge roughness (Supplementary Fig. 10). The width variation and edge roughness of the SI-OLES-based micropatterns exhibited much lower values compared to those of the pristine OLES, indicating that inevitable side-etching phenomenon was effectively suppressed in the SI-OLES. The side damage of the defined patterns

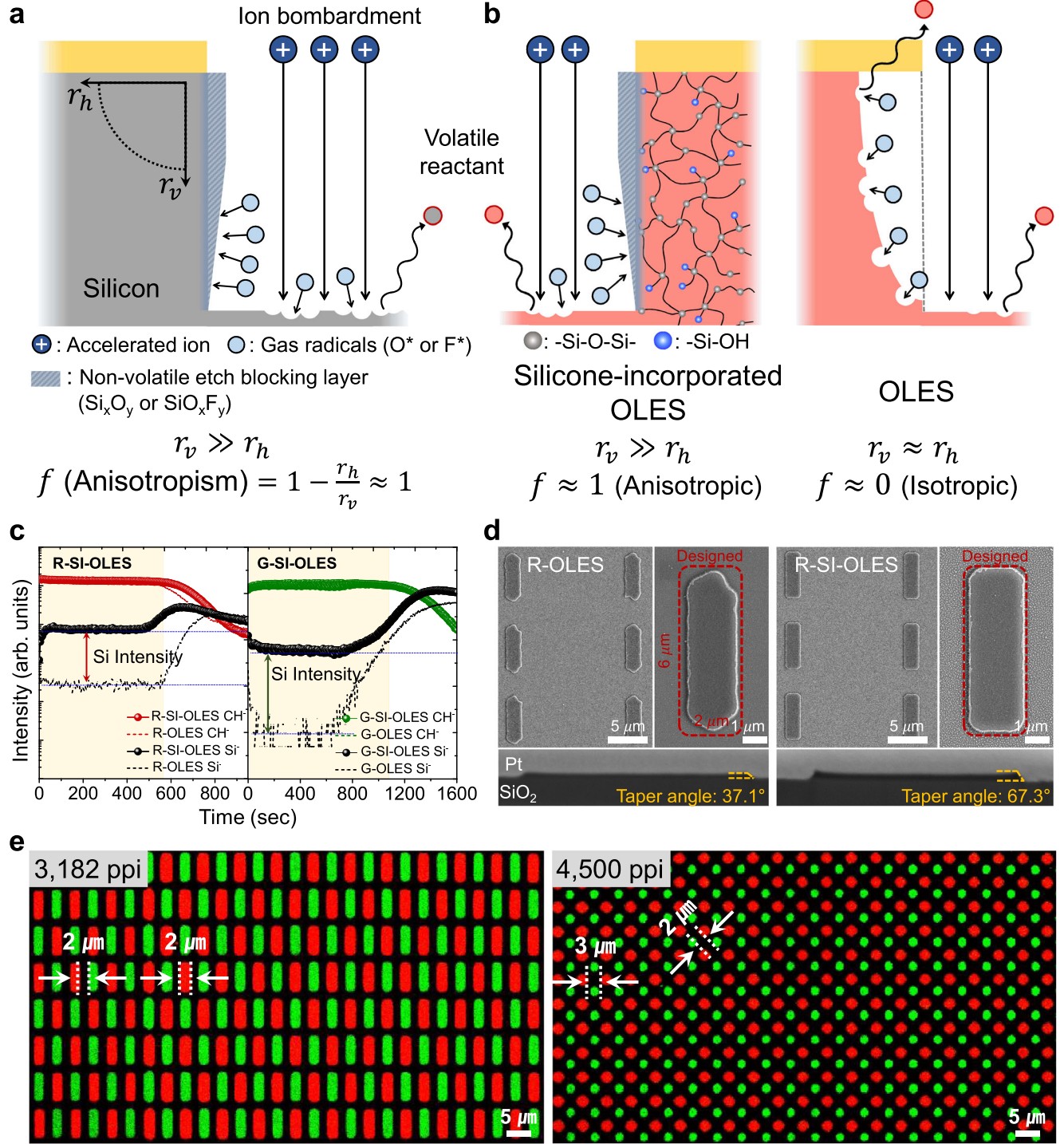

**Fig. 1 | Anisotropic lithography mechanism and characteristics of the SI-OLES.**
**a** RIE-based conventional anisotropic etching mechanism of the silicon (Si).
**b** Conceptual schematic illustration of anisotropic etching behavior induced by the non-volatile EBL in the SI-OLES. **c** ToF-SIMS profiles of the R/G-OLES and R/G-SI-OLES films. **d** Plan (top) and cross-section (bottom) view images of FIB-SEM corresponding to stripe-shaped micropattern arrays of the R-OLES and R-SI-OLES, respectively. **e** Fluorescence microscope images of the ultrahigh-density micro-pixel arrays based on the R/G-SI-OLES.

was also confirmed via second-harmonic generation (SHG) spectroscopy (Supplementary Fig. 11). In the R- and G-OLES patterns, the intensities of the SHG, which are correlated to the degree of physical defection at the side-wall regime of the patterns, were relatively higher than those of the R- and G-SI-OLES. This indicates that the SI-OLES films exhibit greater resistance to the side-etching phenomenon, which can be attributed to the formation of the non-volatile EBL at the sidewall of the SI-OLES patterns. Furthermore, owing to the EBL, etching in the

horizontal direction was significantly alleviated (Supplementary Fig. 12), thereby leading to a significant improvement in the undercut profile of the SI-OLES patterns compared to its counterpart (tapered angle of the micropattern, OLES vs. SI-OLES: 37.1° vs. 67.3°). This indicates that the sidewall EBL of the SI-OLES enables anisotropic lithography during the RCP process. Based on the anisotropic etching behavior of the SI-OLES, R/G micropixel arrays were fabricated through two successive film deposition-RCP cycles (Fig. 1e). The

ultrahigh-density micropixel arrays were achieved with R/G-SI-OLES possessing complicated multicolor pentile patterns with stripe and dot shapes (3182 and 4500 ppi, respectively). It should be noted that anisotropic lithography of the SI-OLES is a vital prerequisite to achieve close-packed pattern arrays with fine pixel pitches, along with physicochemical robustness.

## Formation of non-volatile EBL of SI-OLES films

For the sake of proving the formation of the non-volatile EBL ($Si_xO_y$ or $SiO_xF_y$), which is the core mechanism of anisotropic lithography in the SI-OLES film, the EBL formed on the surface of the R- and G-SI-OLES films was analyzed by X-ray photoelectron spectroscopy (XPS), depending on the exposure to reactive etching gases (Fig. 2a, b and Supplementary Note 2). Before the RIE process was applied, the Si 2$p$ spectra of the R- and G-SI-OLES showed several Si-derivative peaks corresponding to 100.7 eV (Si−C and Si$^+$; Si−O−Si), 101.8 eV (Si$^{2+}$; Si−O), 102.6 eV (Si$^{3+}$; Si$_2$O$_3$), and 103.5 eV (Si$^{4+}$; SiO$_2$), which originated from the silicone networks embedded in the SI-OLES film[28,29]. Intriguingly, after exposure to the reactive etching gases, the high binding energy states (Si$^{3+}$ and Si$^{4+}$) of the Si 2$p$ increased dramatically. This indicates that the silicone blocks on the surface of the SI-OLES film are highly oxidized by reactive gas radicals (O$^•$). This is because the majority of the native silicone components are converted into additional silicon oxide species, which could be acting as the non-volatile EBL. In addition, with respect to F$^•$ gas radicals, the non-volatile EBL ($SiO_xF_y$) was successfully rendered in both the R- and G-SI-OLES films (Supplementary Fig. 13). In this regard, it is noted that the proposed SI-OLES can realize the non-volatile EBL with universal reactive etching gases during RIE process, which is an unexplored phenomenon so far in the OLED technologies.

The non-volatile EBL formed on the surface of the SI-OLES can influence the chemical etching reaction in the vertical direction, depending on the applied etching time. When the OLES films were exposed to reactive etching gases, surface oxidation can be induced by the gas absorption. (Fig. 2c and Supplementary Note 2). It speculates that compared to the OLES films, the EBL derived from the surface of the SI-OLES film could prevent the chemical etching reaction in the vertical direction, resulting in a smaller variation in the initial oxidation level as a function of etching time. As shown in Fig. 2d, e, the initial oxidation level (i.e., increased surface work function by p-doping effect) for the R- and G-OLES films noticeably decreased with the etching time; however, that of the R- and G-SI-OLES films was relatively sustained. This could be attributed to the inhibition of the CH$_2$ scission reaction of the oxidized chain segments by the non-volatile EBL on the surface of the SI-OLES films. In particular, the CH$_2$ scission generated in the surface regime of the OLES films induces a reduction of their molecular weight, so that the surface modulus can significantly decrease[30,31]. As shown in Fig. 2f, g, the initial surface moduli of the OLESs and SI-OLESs were almost identical because the surface regime of the SI-OLES film possesses a polymer-rich phase that is thermodynamically stable in air[32] (Supplementary Fig. 14). The surface moduli of both the R- and G-SI-OLES films were marginally degraded as RIE etching time increased; however, a dramatic reduction in the surface modulus was observed in both the R- and G-OLES films. Along with the aspect of oxidation level alteration of the SI-OLES films, this is a phenomenon caused by the surface-introduced EBL capable of prohibiting the CH$_2$ scission reaction. Note that the above-discussed results so far are key evidence to confirm the formation of the non-volatile EBL for the SI-OLESs during the RIE process.

## EL characteristics of pixelated SI-OLEDs

Not only high-resolution anisotropic lithography of the SI-OLES film using the non-volatile EBL-based RCP process, but also the EL properties and performance of the SI-OLES-based OLEDs should be demonstrated. The R- and G-OLEDs based on the R- and G-SI-OLES

(hereinafter referred to as R- and G-SI-OLED, respectively) were fabricated with a tandem structure of ITO/PEDOT:PSS (30 nm)/R or G-SI-OLES films (60 nm)/Ca (10 nm):Al (150 nm). Both the R- and G-SI-OLEDs exhibited identical or even improved EL performance compared to the pristine R- and G-OLEDs, which implies that the embedded silicone network of the SI-OLES does not interfere with exciton generation or induce quenching within the emission layer (Supplementary Fig. 15). This preserved or improved light-emitting performance of the SI-OLEDs is attributed to the molecular entanglement in the SI-OLES caused by the silicone network, leading to enhanced charge transport within the emission layers[26,27,33–35]. In addition, we fabricated fine-pixelated (10 μm × 10 μm) R- and G-SI-OLED arrays using silicone-engineered anisotropic lithography (through the conventional RCP process) (see Supplementary Table 2 for the detailed luminance efficiency of pristine OLEDs, SI-OLEDs, and micropatterned SI-OLEDs). For the operational stability of the SI-OLED arrays, a pixel-defining layer (PDL) was employed to prevent an undesired electrical short between the cathode and anode (Methods). The R- and G-SI-OLED arrays exhibited clear light emission characteristics despite undergoing micropatterning processes via consecutive RCP processes (Fig. 3a, b and Supplementary Figs. 16, 17). Specifically, the EL spectra of both the pixelated R- and G-SI-OLEDs corresponded exactly to those of the pristine OLEDs and SI-OLEDs, proving the exceptional robustness of the SI-OLES against the harsh lithographic processes (Fig. 3c, d). Furthermore, the current density (J)–voltage (V)–luminance (L) characteristics of the micropixelated R- and G-SI-OLEDs were evaluated (Fig. 3e, f). The V$_{on}$ of the R- and G-SI-OLED arrays possessed comparable values (4.09 and 4.32 V, respectively) to the non-patterned pristine OLEDs and SI-OLEDs; moreover, the L values of both the devices linearly increased without degradation, even under a high operating voltage regime (up to 14 V). Additionally, the L values of the patterned SI-OLEDs were slightly reduced compared to those of the non-patterned SI-OLEDs, which could be attributed to the (i) reduced effective light-emitting area (6 μm × 6 μm) owing to the deposited PDL and (ii) inevitable exposure to O$_2$ and H$_2$O during the RCP process. However, the V$_{on}$ and L characteristics of the patterned SI-OLEDs were comparable to those of previous studies on solution-processable, non-patterned, polymer-based, thermally activated delayed fluorescence OLEDs or perovskite light-emitting diodes[36–38].

## Demonstration of high-resolution, full-color SI-OLEDs

Demonstration of multicolor pixelated SI-OLED arrays using the silicone-engineered anisotropic lithography via consecutive RCP processes is a key route to the realization of ultrahigh-density full-color OLEDs. A micropatterned (15 μm × 15 μm) two-color R/G-SI-OLED array was fabricated via two series of consecutive silicone-engineered anisotropic lithography (Fig. 4a and Supplementary Movie 1). Before applying the RCP process to achieve micropixelation of the second emission layer, an etch stop layer (ESL) was deposited on top of the predefined pixels to prevent unnecessary additional etching reactions (Methods). This ESL was completely removed by using the RIE process after the multicolor patterning fabrication was completed. As shown in Fig. 4b, the EL spectra of the R/G-SI-OLED array were accurately divided into the corresponding spectra of the R- and G-SI-OLEDs. This indicates that the SI-OLES films have a high tolerance against the RCP process. As a proof of concept, for the achievement of high-density RGB OLEDs, solution-processable blue (B)-OLES composed of poly(9,9-dioctyl-9H-fluorene-2,7-diyl) (PFO): poly(9-vinylcarbazole) (PVK): bis(2,4-difluorophenylpyridinato)-tetrakis(1-pyrazolyl) borate iridium(III) (Fir6) was adapted as a final emission layer (Methods and Supplementary Fig. 18). Note that unlike the R- and G-SI-OLES, the B-OLES was not integrated with the silicone network in this study, even though the B-OLES was able to achieve high physico-chemical tolerance when the silicone network was embedded (Supplementary Fig. 19). This is because the B-OLES is based on a ternary blending

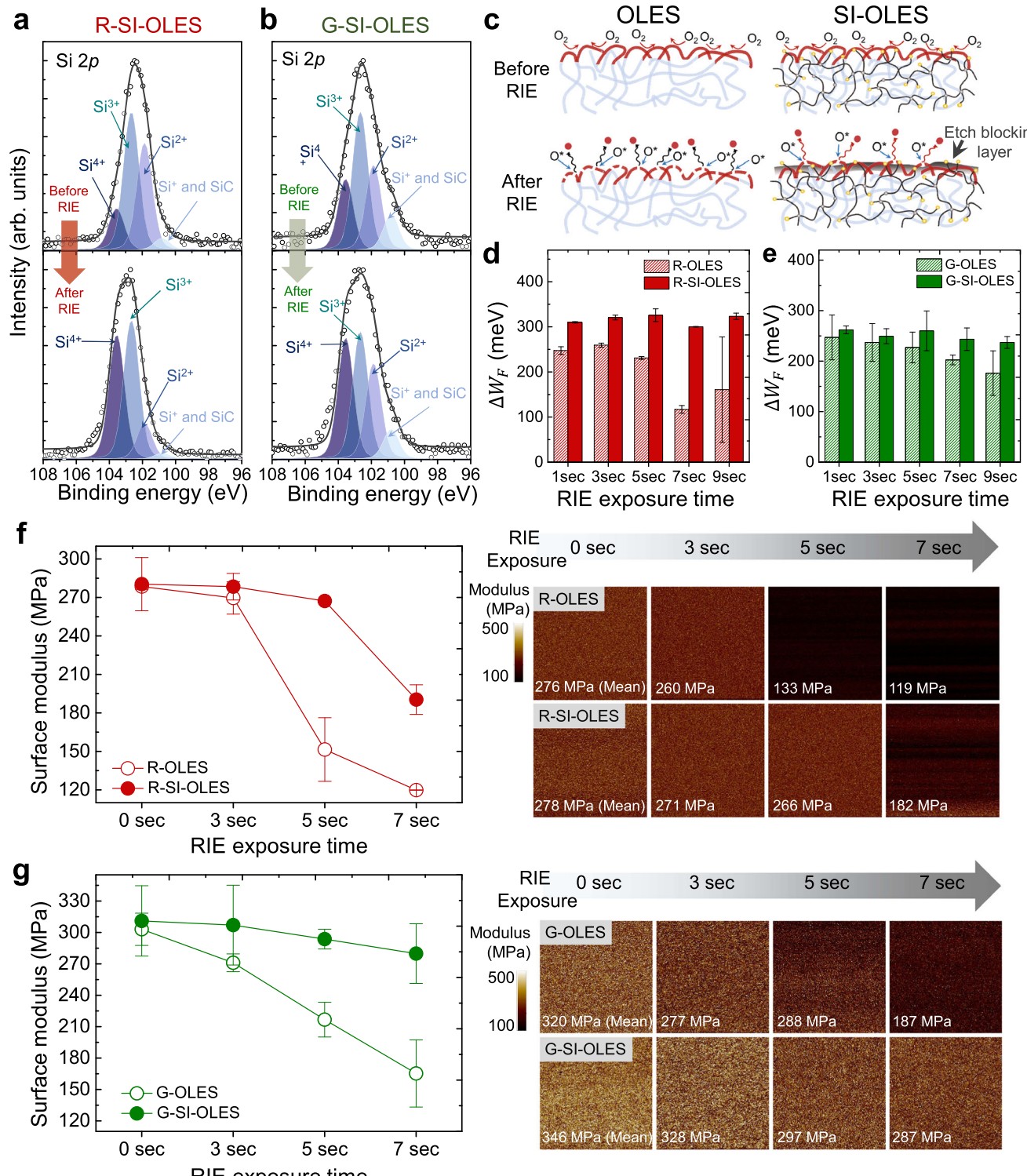

**Fig. 2 | Analysis of the non-volatile EBL of the SI-OLESs. a, b** XPS analysis of the surface atomic structure of the R- and G-SI-OLES based on exposure of reactive etching gases. **c** Schematic illustration of the chemical etching mechanism of the SI-OLES with the non-volatile EBL. **d, e** Variation of increased work functions of the R- and G-OLES/SI-OLES films as a function of reactive etching times, respectively. Mean values with ± standard variations are indicated in the data. **f, g** Surface modulus values and mapping images of the R- and G-OLES/SI-OLES films in terms of reactive etching times, respectively. Mean values with ± standard variations are indicated in the data.

system (host-dopant-hole transporting materials) to enhance its luminescence efficiency; however, the incorporated silicone network seemed to interfere energy transfer process within the B-OLES layer, so that the light-emitting performance significantly deteriorated. These are why the B-OLES was directly utilized even though the

micropatterned feature was relatively distorted compared to the R- and G-SI-OLES (Supplementary Fig. 20). We anticipate that the development of high-efficient B-OLES that is not required of doping and buffer materials would provide a reasonable solution to realize the B-SI-OLES. Micropatterned two-color OLED arrays of the R- and G-SI-OLES films

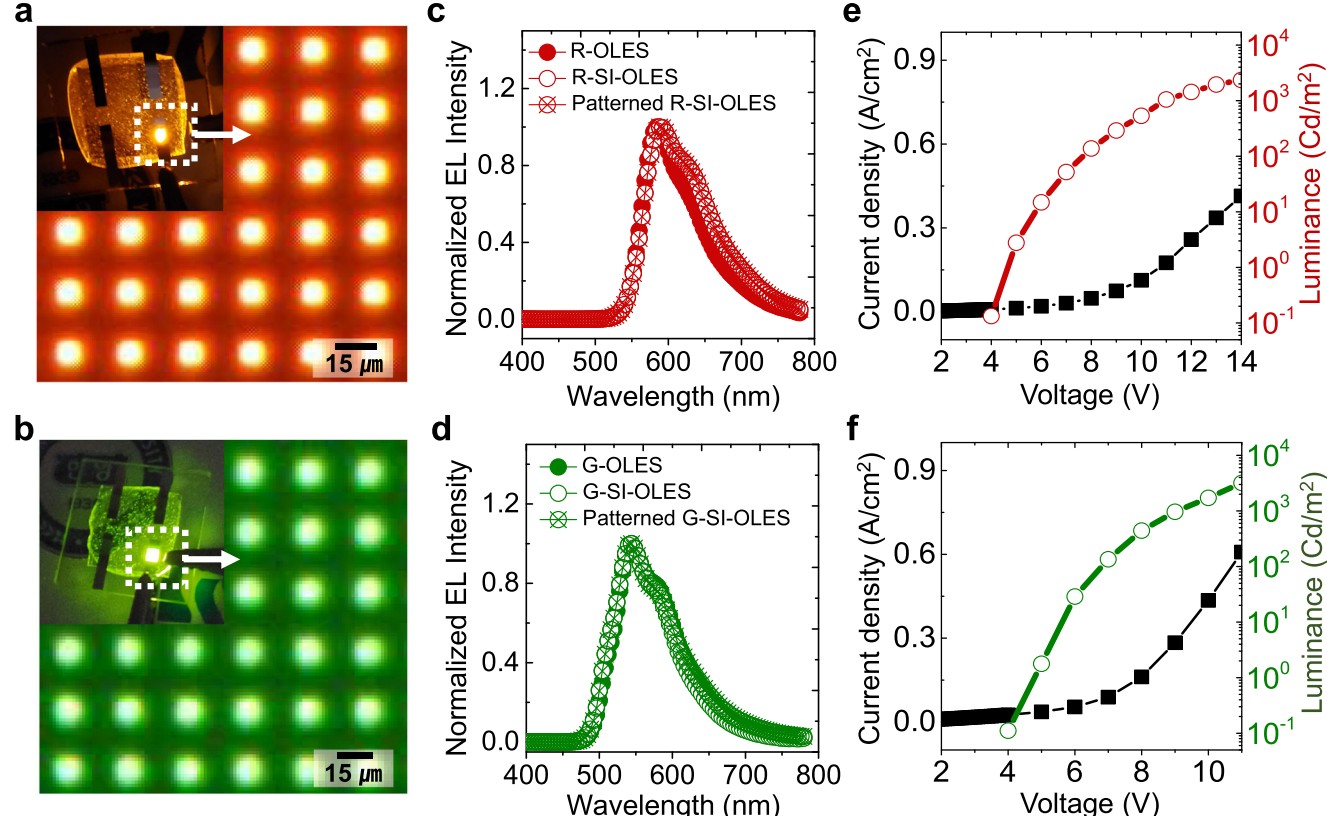

**Fig. 3 | EL characteristics of high-resolution pixelated R- and G-SI-OLEDs.**
**a**, **b** Optical microscopy and photography images (inset) of electrically operating R- and G-SI-OLEDs. **c**, **d** EL spectra of the R- and G-SI-OLESs before and after pattern fabrications. **e**, **f** Current density (J)-voltage (V)-luminance (L) characteristics of the micropixelated R- and G-SI-OLEDs.

combined with the B-OLES were implemented through the RCP process (Fig. 4a, b and Supplementary Figs. 21, 22).

Notably, provided that the R- and G-SI-OLES films were inherently prone to the RCP-based patterning method, multicolor OLED arrays would not have been realized by the consecutive RCP processes. Consequently, we fabricated an ultrahigh-density RGB pixel array corresponding to a sophisticated stained-glass illustration as well as an ultrafine-pixelated full-color SI-OLED array via the application of three cycles of consecutive RCP processes (Supplementary Fig. 23). The fluorescence image of the stained-glass pattern exhibited definite color representation by combinations of delicate RGB micropixel array (4216 ppi) (Fig. 4c). The obvious color expression is attributed to extremely dense RGB pixels, which is corresponding to 4,938,271 patterns/cm². Note that even though the full-color OLES pattern array was fabricated, it showed an extremely higher pattern density than previously reported mono- or two-color OLES pattern arrays[39–44] (Fig. 4d). Thanks to this superior patterning capability of the SI-OLES, furthermore, the full-color SI-OLED array corresponding to 949 ppi was successfully achieved, exhibiting the highest resolution compared to the reported full-color OLED arrays via various OLED patterning methods[45–55] (Fig. 4e, f and Supplementary Fig. 24). This indicates that the RCP process has the best capability to realize ultrahigh-density OLED arrays, and that the SI-OLES is a promising solution to apply the RCP-based micropatterning method to high-resolution OLED technologies.

In summary, the proposed SI-OLES material design, inspired by silicon electronic technology, enables enhanced anisotropic etching behavior as well as high chemical tolerance, thereby resulting in ultrahigh-resolution patterning with high-throughput and high-fidelity via the RCP method into the OLEDs. We systemically demonstrated silicone-engineered anisotropic lithography originating from the in-situ formation of the non-volatile EBL of the SI-OLES during the RIE. Based on this mechanism, the ultrahigh-density (4500 ppi) SI-OLES pattern array was achieved using the RCP process. Finally, we successfully fabricated the high-density full-color SI-OLEDs via successive RCP-based micropatterning, which is typically not applicable to conventional OLED materials. The proposed approach can be extended to organic electronic applications requiring high resolution, such as organic image sensors, neuromorphic electronics, and NeuroGrid technology[56–58].

## Methods

### Materials

Processing solvents, including chloroform (CF) and chlorobenzene (CB), were purchased from Sigma-Aldrich and were used as received. Silicone network precursor, 1,8-bis(trichlorosilyl) octane (≥97%), was purchased from Gelest and was also used as received. Poly(3,4-ethylenedioxythiophene):poly(styrenesulfonate) (PEDOT:PSS, AI4083) was purchased from Ossila. Poly [2-methoxy −5-(2′-ethylhexyloxy)-*p*-phenylene vinylene] (MEH-PPV; $M_w = 80,000$), poly(9,9-dioctylfluorene-co-bithiophene) (F8T2; $M_w = 63,525$), and poly (9-vinylcarbazole) (PVK; $M_w = 25,000–50,000$), poly(9,9-di-*n*-octylfluorenyl-2,7-diyl) (PFO; $M_w = 50,000–150,000$), and bis(2,4-difluorophenylpyridinato)-tetrakis(1-pyrazolyl) borate iridium (III) (Fir6; $M_w = 851.65$), which are R-, G-, and B-OLES respectively, were purchased from Ossila and Sigma-Aldrich. Parylene-C, which is acting as a pixel define layer (PDL) and etch stop layer (ESL), was purchased from Daisan Kasei Co., Ltd. Photoresist (AZ-5214E) and developer (AZ 300MIF) were purchased from AZ Electronic Materials and Merck Performance Materials, respectively. The Ca (≥99.9%) and Al (≥99.999%) were used as cathode.

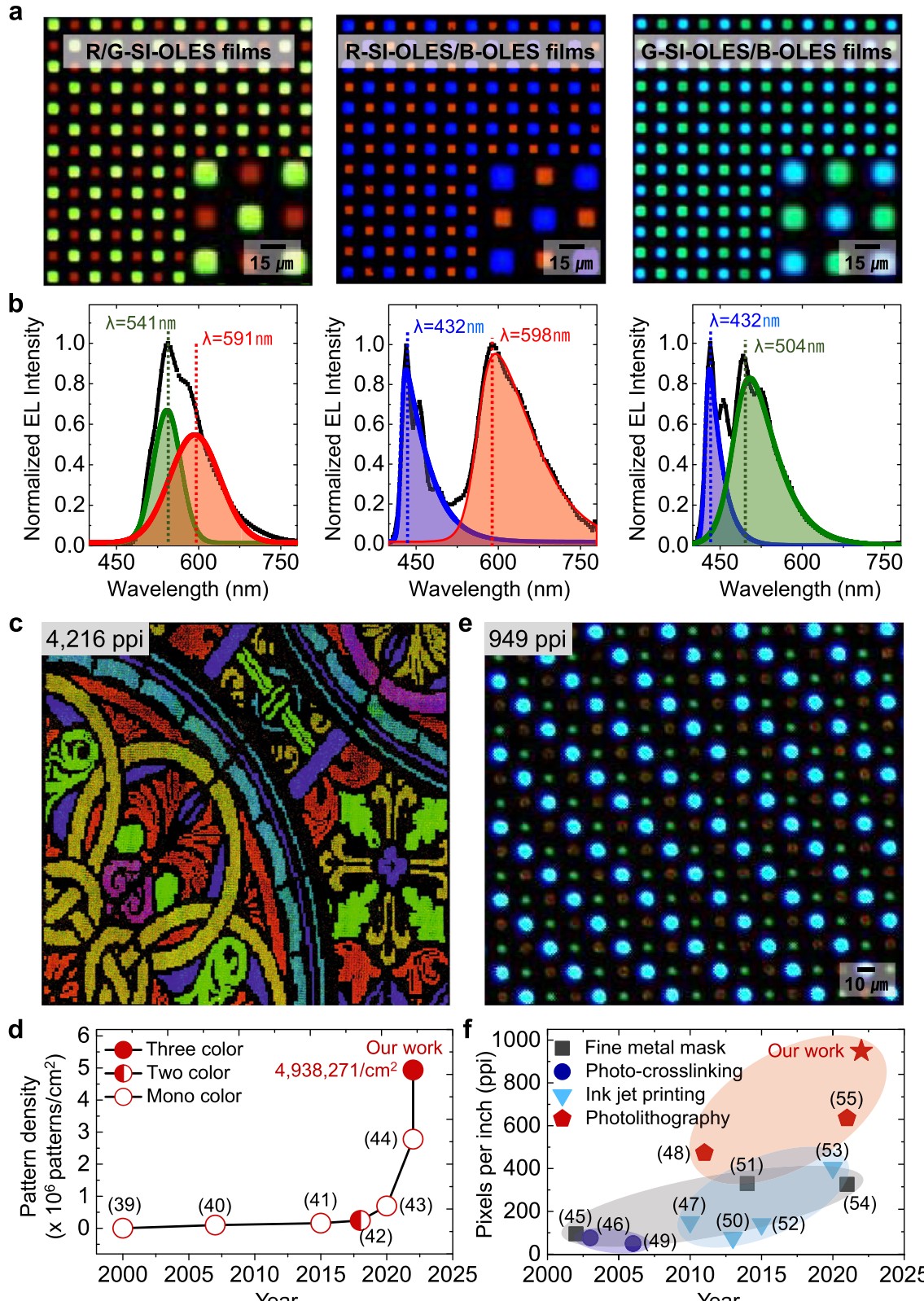

**Fig. 4 | EL characteristics of multicolor pixelated, high-resolution SI-OLEDs using silicone-engineered anisotropic lithography (via consecutive RCP processes). a, b** Optical microscopy images (**a**) and EL spectra (**b**) of electrically operating multicolor pixelated SI-OLED arrays corresponding to the R/G-SI-OLES (left), R-SI-OLES/B-OLES (middle), and G-SI-OLES/B-OLES (right). **c** Fluorescence stained-glass image of a micropatterned array (4216 ppi) composed of the R/G-SI-OLES and blue-based micropixels (size of 3 μm × 3 μm with 1.5 μm of pitch). The cyan, yellow, and magenta colors correspond to a combination of G/B, R/G, and R/B pixels, respectively. **d** The progress of the demonstrated pattern density of mono-, two-, or three-color OLES arrays in the past decades (refs. 39–44). **e** Optical microscopy image of electrically operating ultrahigh-density (949 ppi) of the full-color SI-OLED array. **f** Comparisons of the previously reported resolution of full-color OLED arrays with our work (refs. 45–55).

## Preparation and characterization of the SI-OLES film

A solution for the R-SI-OLES films was prepared by dissolving the MEH-PPV and silicone network precursors with the designed volume ratio in CB. The 0.4 ml of MEH-PPV solution (6 mg/ml in CB) and the silicone network precursors (0.2 µl) were mixed at 80 °C for 1 h. After that, the solution was spin-coated at 1000 rpm for 1 min onto glass or Si/SiO$_2$ substrate, and then the resulting film was annealed at 180 °C for 3 h in a nitrogen environment. For the G-SI-OLES film, the 0.4 ml of the F8T2 solution (7 mg/ml in CB) and the silicone network precursors (0.5 µl) were mixed at 80 °C for 1 h. The solution was spin-coated at 1000 rpm for 1 min onto glass or Si/SiO$_2$ substrate, and then the resulting film was annealed at 180 °C for 12 h in a nitrogen environment. Finally, a solution for the blue film was prepared by dissolving the PVK, PFO, and Fir6 in CF, followed by stirring at 80 °C for more than 12 h, respectively. The PFO and PVK (1:1) were mixed and added with Fir6 (1 v/v%). The blending solution was spin-coated at 1000 rpm for 1 min onto glass or Si/SiO$_2$ substrate. The thickness of the films was measured by a surface profilometer (Bruker, Dektak XT-E) and AFM (Park systems, NX10). UV absorption and fluorescence spectra of OLES films were measured by a UV–Vis–NIR spectrophotometer (Jasco, V-770) and fluorescence spectrometer (Scinco, FS-2), respectively. AFM images were obtained under ambient conditions in a tapping mode. The measurement of photoluminescence images was conducted with a Confocal Laser Scanning Microscope system from SP8 X with excitation wavelength 458–580 nm. Time-of-flight secondary ion mass spectrometry (ToF-SIMS) analysis was performed with ToF-SIMS-5 (ION-ToF, Germany) instrument. Negative ion spectra were obtained from the given sample area (100 µm × 100 µm) with the primary ion source (Bi$_3^{2+}$; 30 keV), and sputter etching was carried out with an Ar1000$^+$ beam (2.4 keV). The top and cross-sectional SEM images of the OLES films were obtained by focused ion beam-field emission scanning electron microscope (FIB-FESEM, Scios) at Hanyang Center for Research Facilities (Seoul). X-ray photoelectron spectroscopy (XPS, ESCA Veresprobe II) was conducted using a monochromatic Al Kα source ($hv$ = 1486.7 eV) with a pass energy of 29.5 eV. Second-harmonic generation (SHG) spectroscopy were performed using 390 nm (2ω) harmonics generated by a 790 nm (ω) of a femtosecond laser incident on pattern array samples. The kelvin probe force microscopy (KPFM) and nanomechanical analysis were conducted by Park NX10. The KPFM analysis was carried out with probes from Park Systems (NCS36/Cr-Au, resonant frequency = ~65 kHz, k (spring constant) = ~0.6 N/m, tip radius ~35 nm), and the surface modulus and mapping images were obtained with probes from Park Systems (SD-R30-FM, resonant frequency =~75 kHz, k (spring constant) =~2.8 N/m, tip radius ~30 nm).

## Patterning of the SI-OLES film

The SI-OLES films were patterned by conventional photolithography (AZ-5214E photoresist and AZ 300MIF developer) and reactive ion etching (RIE, Oxford Instruments PlasmaPro 80) process. The photoresist was spin-coated at 4000 rpm for 1 min on the pre-deposited SI-OLES films, and then soft-baking was conducted at 100 °C for 90 s. The photoresist-coated SI-OLES film was exposed to the UV source (365 nm, 25 mW/cm$^2$) for 10 sec with a photomask using a mask aligner (Karl Suss MA-6 Mask Aligner), and the photoresist pattern was developed by washing with the developer for 40 s at room temperature. The photoresist-patterned SI-OLES films were exposed to Ar/O$_2$ (40 sccm/20 sccm) or CF$_4$ (60 sccm) etching gases for 60–90 s with 50 W radio-frequency power under a pressure of 0.05 torr. After that, the patterned photoresist was removed by sonication process with acetone and isopropyl alcohol, so that micropatterns of the SI-OLES were produced.

## Fabrication and performance measurement of SI-OLEDs

A glass substrate with a pre-patterned indium-tin-oxide (ITO) electrode (~20 Ohm/sq) was used for the OLED fabrication. The glass substrate was sonicated with acetone, isopropyl alcohol, and de-ionized water

sequentially for 15 min each. The cleaned substrate was treated with UV/O$_3$ for 20 min. Parylene-C as a PDL was deposited on the substrate, and then through the RIE-coupled photolithography (RCP) process (deposition of photoresist pattern−reactive ion etching process−removal of photoresist pattern), PDL possessing a groove with 10 µm × 10 µm × 150 nm was fabricated. After that, PEDOT:PSS (AI4083) as a hole injection layer was spin-coated onto the ITO-patterned glass substrate at 4000 rpm for 30 sec and the resulting film was annealed at 180 °C for 30 min in ambient condition. The first emission layer (R-SI-OLES) was spin-coated on the PDL-deposited substrate and annealed at 180 °C for 3 h in a nitrogen atmosphere. Then, a Parylene-C film (50 nm) or Al$_2$O$_3$ serving as the ESL was deposited. The R-SI-OLES layer with the ESL was patterned with desired pixel size through the RCP process. Also, the second emission layer (G-SI-OLES) was formed by spin-coating on as-fabricated R-SI-OLES pattern arrays and annealed at 180 °C for 12 h. The second ESL was deposited on the G-SI-OLES layer, and the G-SI-OLES layer with the ESL was pixelated through the RCP process. Finally, the third emission layer (B-OLES film) was spin-coated and annealed at 180 °C for 1 h, which was patterned by the RCP processes. Prior to the removal of the photoresist on the pixelated B-OLES layer, the ESL layers, deposited on the R and G-SI-OLES pattern array, respectively, were completely eliminated by an additional RIE process. After the photoresist strip process of B-OLES patterns, the full-color pixelated SI-OLES arrays was achieved. The Ca (10 nm)/Al (150 nm) as a cathode electrode was thermally evaporated at high vacuum conditions (2.0 × 10$^{-6}$ Torr). The EL performance of the SI-OLEDs was evaluated by a spectro-radiometer (PR 655) while the input voltage was swept from 0 V to 12 V using a Keithley 2400.

## Data availability

All data were included in this article and its supplementary information files. All data were available from the corresponding authors upon reasonable request.

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

## Acknowledgements

This work was supported by National R&D Program through the National Research Foundation of Korea (NRF), funded by the Ministry of Science and ICT (2021M3D1A2049315). This work was also supported by the Basic Science Research Program of the National Research Foundation of Korea (NRF), funded by the Ministry of Science and ICT (2020R1A2C3014237).

## Author contributions

H.L. and D.H.K. supervised the project. H.K. and D.H.K. conceived the concept. H.K., K.-Y.C., and H.W.P. designed and carried out experiments. H.K., H.W.P., U.J., B.H. and D.H.K. analyzed and interpreted material characteristics. K.-Y.C., R.L., and H.L. fabricated and evaluated OLEDs. M.J.K., H.H., and K.-B.C. performed XPS (X-ray photoelectron spectroscopy) and SHG (Second-harmonic generation) analysis. S.L. and J.-Y.K. fabricated an $Al_2O_3$ layer as an etch stop layer. M.S.K. discussed the results and commented on the manuscript. H.K., K.-Y.C., H.W.P., H.L., and D.H.K. co-wrote the paper.

## Competing interests

The authors declare no competing interests.
