## [Peer Review File · Nature Communications]

REVIEWER COMMENTS

Reviewer #1 (Remarks to the Author):

This work by Choi et al. demonstrates ultra-high-density OLEDs through novel engineered lithography technique. The work is well performed, has a methodical flow and research gaps have been underlined with clarity. However, in the current form I have reservations on publication of this work due to the poor OLED performance and lack of confirmation of this technique for universal applicability for OLEDs materials. The pixelated OLEDs for all the colors perform quite poorly and authors have not represented the efficiency metrics of the devices. Further, while RGB OLEDs are good for the proof of concept, the device structure and OLES materials used in these devices are outdated. I insist that authors show further confirmation for robustness of their lithography technique by applying it to vacuum deposited OLEDs materials. I believe compatibility of the technique to multilayered OLED stack will be useful. Lastly, current commercial standards of OLED technology are achieved with TADF and phosphorescent materials. Will this be compatible with 2nd and 3rd generation OLEDs materials? What is the limitation of proposed lithography technique?

Reviewer #2 (Remarks to the Author):

The authors introduced a high-resolution patterning of light-emitting polymer using silicon-engineered lithography for the fabrication of organic light-emitting diodes (OLEDs). It provides strong anisotropic etching by using a non-volatile etch-blocking layer (EBL) that in-situ forms. It leads ultrahigh density OLED pattern with reactive ion etching (RIE)-coupled photolithography (RCP) process. The authors finally fabricate full-color RGB OLEDs using micropatterning. Overall, the idea is novel, and the manuscript is well-written. Also, there are enough characterization and discussion for the experimental results. Therefore, I think this manuscript can be suitable for publication in Nature Communications after thorough minor revisions. Below are suggestions that might help to improve the paper before it is published.

1. The authors claimed that their silicone-incorporated organic light-emitting semiconductors (SI-OLES) have strong anisotropic etching profiles like silicon due to EBL on the sidewall. Even though there are many characterization results to prove the mechanism of anisotropic lithography, it is hard to confirm the impact of EBL in micropatterning except SEM images of one rectangle-shaped micropattern in Fig. 1d and Fig. S7.

It is better to show a quantitative comparison of horizontal etch rate and vertical etch rate between SI-OLES and control polymers.

Also, for discussion on the tapered angle of the micropattern, it is better to show more images to prove their high uniformity and yield for micropatterning.

2. The SI-OLES showed even improved EL performance compared to the pristine polymers. Why did they show the improvement? It is better to add the reason and related discussion.

3. When I checked the micropatterns of blue layers which do not have silicone-engineering in Fig. S17 – 19, the patterning results were not so different with red and green SI-OLES layers. It would be interesting to do supplementary experiments or characterization to emphasize the strong anisotropic etching in the manuscript.

Reviewer #3 (Remarks to the Author):

The manuscript entitled "Silicone engineered anisotropic lithography for ultrahigh-density OLEDs" submitted by D. H. Kim and coworkers presented intriguing etching characteristics of silicone-incorporated organic light-emitting semiconductors (SI-OLES), enabling to be delicately pixelated by conventional photolithography and reactive ion etching. The approach that unique etching behavior of silicon is implanted on the OLED materials is particularly noteworthy, and this technique seems to significantly contribute improvement of pattern resolution in the field of the OLEDs. Also, the authors systematically described the non-volatile blocking layer that is induced during etching of the SI-OLES, as well as the developed high density RGB OLED pattern arrays and the light-emitting performance were quite impressive. In addition, the manuscript is well-written, and the figures are well-organized. I recommend this high quality manuscript for publication in Nature Communications after minor revision, and several points should be addressed before publication.

1. The authors mentioned that for analysis of non-volatile etch-blocking layer generated on the surface regime of the SI-OLES film, the lowest RF power of RIE was deliberately used. However, as the RF power is too low, the etching performance of RIE can be extremely reduced because ion bombardment effect is not efficiently developed. How can it confirm that the variation of surface modulus of the SI-OLES is the results from the etching reaction of RIE? Also, the authors should indicate the detailed RIE condition.

2. In the EL performance of OLEDs based on the OLES and SI-OLES, both R and G-SI-OLES showed higher current density than the pristine ones. This seems to importantly affect the preservation or improvement of luminous property of the SI-OLEDs, compared to pristine OLEDs. What is the origin of the enhanced electrical characteristics of the SI-OLES even though the insulating materials are embedded?

3. Does the measured contact angle of the SI-OLES film correspond to either R-SI-OLES or G-SI-OLES? This is not clear in the manuscript.

4. The author mentioned that the used Blue-OLES in this work was not incorporated with silicone network because the luminescence performance of the SI-B-OLES was too low, and this problem can be solved if high efficient Blue-OLES is exploited. However, even if high efficient blue materials are developed, it could be meaningless if the chemical robustness of bluish semiconducting materials is not demonstrated when silicone network is adapted. At least, the authors should provide additional data that even blue materials can possess high tolerance when silicone network is integrated.

5. Several typos should be corrected.

Reply to Reviewer #1

This work by Choi et al. demonstrates ultra-high-density OLEDs through novel engineered lithography technique. The work is well performed, has a methodical flow and research gaps have been underlined with clarity. However, in the current form I have reservations on publication of this work due to the poor OLED performance and lack of confirmation of this technique for universal applicability for OLEDs materials.

Response: We thank the reviewer for the invaluable comments. We revised the manuscript according to the reviewer's comments.

Q1) *The pixelated OLEDs for all the colors perform quite poorly and authors have not represented the efficiency metrics of the devices.*

Response: We appreciate the reviewer's comment. We understand the reviewer's concern related to the device performance. As the reviewer commented, light-emitting characteristics of pristine OLEDs, SI-OLEDs, and micro-patterned SI-OLEDs are summarized in **Supplementary Table R1**. Our SI-OLEDs might be difficult to utilize as well-defined high efficiency displays. The luminance efficiency is, however, comparable or even better to recently presented, non-patterned light-emitting polymer-based OLEDs (*Adv. Mater.* **2022**, *34*, 2201844; *Sci. Adv.* **2021**, *7*, eabd9715; *Angew. Chem. Int. Ed.* **2021**, *60*, 7220-7226), even when our SI-OLEDs underwent micro-patterning fabrication. In this regard, we note that although the SI-OLEDs are micro-pixelated, the light-emitting performance is not inferior to that of the non-patterned OLEDs.

Furthermore, currently commercialized OLED displays exhibit maximum luminance values of 500 to 1000 cd/m² (Specifically, high dynamic range (HDR) of OLED TVs and iPhone 14 OLED displays are 1,000 and 800 cd/m², respectively) (*Adv. Funct. Mater.* **2021**, *31*, 2009336; *Opt. Express* **2017**, *25*, 33643-33656), which is the level that the demonstrated SI-OLEDs can sufficiently satisfy in terms of maximum luminance. Therefore, we think that the light-emitting capability of the SI-OLEDs can be reasonably acceptable in the field of OLEDs, and the SI-OLEDs possess considerable potential for practical usage.

To facilitate the readers to figure out the OLED characteristics clearly, we added this table as **Supplementary Table 2** in the revised manuscript, which obviously shows the efficiency metrics of the SI-OLEDs.

Supplementary Table R1 | Summary of light-emitting performance of the OLEDs based on pristine OLES, SI-OLES, and micro-patterned SI-OLES.

Materials	Maximum luminance	Maximum current efficiency	Turn-on voltage	EL peak wavelength
-----------	-------------------	----------------------------	-----------------	--------------------

	[cd/m ²]	[cd/A]	[V]	[nm]
R-OLES	5270	0.28	2.03	584
R-SI-OLES	5619	0.28	2.05	588
Micro-patterned R-SI-OLES	2349	0.58	4.32	588
G-OLES	4940	0.42	2.54	544
G-SI-OLES	5303	0.46	3.01	544
Micro-patterned G-SI-OLES	3186	0.52	4.53	544

Q2) Further, while RGB OLEDs are good for the proof of concept, the device structure and OLES materials used in these devices are outdated. I insist that authors show further confirmation for robustness of their lithography technique by applying it to vacuum deposited OLEDs materials. I believe compatibility of the technique to multilayered OLED stack will be useful.

Response: We appreciate the critical comments from the reviewer. As the reviewer mentioned, vacuum-processable OLES (mostly phosphorescent small molecules) is currently leading commercialized OLED displays. On the other hand, solution-processable OLESs (small molecules and polymers) have been regarded as underdogs due to their low reliability and luminance efficiency as well as premature patterning processes applicable to these materials.

Thanks to recent advancement of synthesizing and processing techniques of solution-processable materials, solution-processed OLEDs are no longer behind on vacuum-processed OLEDs in terms of uniformity and light-emitting performance (*Adv. Funct. Mater.* **2019**, 29, 1901025; *Nat. Commun.* **2015**, 5, 5756). In particular, polymer LEDs have significantly gained much attention in high-performance flexible or stretchable OLEDs (*Nature* **2022**, 603, 624-630; *Adv. Mater.* **2022**, 34, 2201844; *Sci. Adv.* **2021**, 7, eabd9715). In this regard, we consider that the used OLES materials in this work are not outdated.

While we believe that confirming the applicability of our strategy to fairly different materials systems (vacuum deposited small molecules vs. solution processed polymers) is beyond the scope of the current manuscript, we agree that strong impact can be made if it can be exploited to pattern vacuum deposited OLEDs via photolithography. In fact, as an obvious next step, we are currently investigating the silicone engineered anisotropic lithography on phosphorescent small-molecular host-dopant system (see below for details in the reviewer's following comment). We hope this following work can be finalized in the near future.

The suggested multilayered architecture would be beneficial for the enhancement of luminance efficiency of the SI-OLEDs (*Nature* **2022**, 603, 624-630; *Nat. Commun.* **2015**, 5, 5756). As the reviewer pointed out, we emphasize that multilayered SI-OLEDs can be effortlessly achieved, because the chemical

robustness of the SI-OLES is directly exploited through successive film deposition processes to form multilayers. Absolutely, the multilayered structure of the SI-OLEDs should be implemented to optimize their light-emitting performance, and this will be carried out near future study.

Despite all the promises, we believe that the scientific novelty and main point of this manuscript are to present a new concept of silicone engineered anisotropic lithography of OLEDs and systemically investigate the non-volatile EBL-induced etching mechanism.

Q3) *Lastly, current commercial standards of OLED technology are achieved with TADF and phosphorescent materials. Will this be compatible with 2nd and 3rd generation OLEDs materials? What is the limitation of proposed lithography technique?*

Response: We appreciate the reviewer's comments. As mentioned the above, we completely understand a need to extend the silicone engineered anisotropic lithography to universal OLED materials, such as phosphorescent small molecule materials, for convincing its versatility. However, the proposed ladder-like silicone network could not be available to conventional small molecule materials because it is hard to render 3-dimensional molecular entanglement with the small molecules. To overcome this limitation, with cooperation of OLED material synthesis experts, we are developing a new type of solution-processable phosphorescent small molecules (host-dopant system as *2nd and 3rd generation OLEDs materials*) that can be directly linked to silicone networks (Please note that detailed molecular structures cannot be described due to confidential issues). Based on the small materials, we confirmed that not only physico-chemical robustness can be obtained even in the small molecule system, but also fine micropatterns can be achieved by reactive ion etching (RIE)-coupled photolithography (**Figure R1**). Therefore, we carefully appeal that the silicone engineered anisotropic lithography can be effectively adaptable to phosphorescent small molecules considered as standard OLED materials, which will be addressed in our further work.

Figure R1 | Evaluation of physico-chemical robustness of silicone-incorporated phosphorescent small molecule materials. a, Film retention after solvent rinsing and etch rate of SI-phosphorescent materials. **b,** Fluorescence microscope image of micropatterns of SI-phosphorescent materials fabricated by reactive ion etching-coupled photolithography.

Reply to Reviewer #2

The authors introduced a high-resolution patterning of light-emitting polymer using silicon-engineered lithography for the fabrication of organic light-emitting diodes (OLEDs). It provides strong anisotropic etching by using a non-volatile etch-blocking layer (EBL) that in-situ forms. It leads ultrahigh density OLED pattern with reactive ion etching (RIE)-coupled photolithography (RCP) process. The authors finally fabricate full-color RGB OLEDs using micropatterning. Overall, the idea is novel, and the manuscript is well-written. Also, there are enough characterization and discussion for the experimental results. Therefore, I think this manuscript can be suitable for publication in Nature Communications after thorough minor revisions. Below are suggestions that might help to improve the paper before it is published.

Response: We appreciate the positive and constructive comments from the reviewer. We revised the manuscript according to the reviewer's comments.

Q1) *The authors claimed that their silicone-incorporated organic light-emitting semiconductors (SI-OLES) have strong anisotropic etching profiles like silicon due to EBL on the sidewall. Even though there are many characterization results to prove the mechanism of anisotropic lithography, it is hard to confirm the impact of EBL in micropatterning except SEM images of one rectangle-shaped micropattern in Fig. 1d and Fig. S7. It is better to show a quantitative comparison of horizontal etch rate and vertical etch rate between SI-OLES and control polymers. Also, for discussion on the tapered angle of the micropattern, it is better to show more images to prove their high uniformity and yield for micropatterning.*

Response: We thank the reviewer's comments. We totally agree that to prove the concept of silicone-induced anisotropic etching behavior of the SI-OLES more clearly, additional analysis and quantitative assessment of the resulted anisotropic patterns should be discussed.

Figure R2 | Determination of horizontal etch rate. **a**, Schematic illustration of side etching behavior induced by horizontal etching profiles. **b**, Calculation of horizontal etch rates of OLES and SI-OLES.

Development of anisotropic etching profiles indicates that the horizontal etching behavior is significantly suppressed compared to the vertical one, resulting in considerable reduction of inevitable side-etching properties underneath defined photoresist-patterns (**Figure R2a**). By utilizing this phenomenon, horizontal etch rates can be approximately calculated by evaluating the difference of width between the desired and the real patterns. As shown in **Figure R2b**, the estimated horizontal etch rates of the R- and G-SI-OLES decreased by 61.6 and 61.87 %, respectively, compared to that of R- and G-OLES. We note that the reduction of horizontal etching behavior is the key evidence of the augmented anisotropism of the SI-OLES (**Figure 1a,b**).

Figure R3 | Evaluation of pattern fidelity of micropatterns. **a**, Conceptual illustration of pattern width variation and edge roughness. **b,c**, Evaluation of the key parameters of pattern fidelity corresponding to the OLES and SI-OLES-based micropatterns ($2 \mu\text{m} \times 6 \mu\text{m}$).

Moreover, this enhanced anisotropic etching behavior can greatly increase a pattern fidelity due to effective alleviation of the side-etching profiles which induce undesired variation of pattern width and severe distortion of pattern edge roughness (**Figure R3a**). As shown in **Figure R3b and R3c**, the pattern width variation and edge roughness of R- and G-SI-OLES were significantly improved. This indicates that

the SI-OLES can achieve more reliable fine patterns by exploiting its anisotropic etching behavior than the pristine OLES. In conjunction with the evaluation of the pattern fidelity, second-harmonic generation (SHG) spectroscopy is a powerful tool to confirm the degree of physical defection on the side-wall regime of the patterns. The non-linear SHG signal intensities of the R- and G-OLES patterns were higher than that of the R- and G-SI-OLES (**Supplementary Fig. 11**), implying that high uniformity in the side-wall regime of the SI-OLES patterns was obtained. Consequently, we believe that the systemically conducted investigations and in-depth discussions are sufficient for the confirmation of the anisotropic etching behavior of the SI-OLES.

To more clearly reveal and elucidate the anisotropic etching behavior of the SI-OLES, we added the **Figure R2** and **Figure R3** as **Supplementary Figure 12** and **Supplementary Figure 10**, respectively, and the following sentence on the page 6 in the revised manuscript.

“To investigate the precision of the SI-OLES-based micropatterns in details, we quantitatively evaluated the key assessment factors of pattern fidelity, such as variation of pattern width and degree of pattern edge roughness (Supplementary Fig. 10). The width variation and edge roughness of the SI-OLES-based micropatterns exhibited much lower values compared to those of the pristine OLES, indicating that inevitable side-etching phenomenon was effectively suppressed in the SI-OLES.”

Q2) *The SI-OLES showed even improved EL performance compared to the pristine polymers. Why did they show the improvement? It is better to add the reason and related discussion.*

Response: We appreciate the reviewer’s comment. An initial mixing solution of the polymers and silicone network precursors shows homogenous phase. However, as the ladder-like silicone network is being constructed by sol-gel reaction, molecular micro-phase separation between the polymer chains and the assembling network occurs, so that thermodynamic molecular entanglement is spontaneously induced (**Supplementary Fig. 1**). As a result, 3-dimensional molecular entanglement between the silicone network and the polymer chains is achieved, indicating that the polymer chains are molecularly aggregated. The induced chain aggregation facilitates charge transporting capability, so that the electrical characteristics of the polymer semiconductor are not deteriorated even though insulating materials are introduced (*Adv. Mater.* **2019**, *31*, 1901400; *ACS Appl. Mater. Inter.* **2020**, *12*, 55107-55115). Recently, the molecular aggregation mechanism has been exploited in the field of polymer LEDs (*Nature* **2022**, *603*, 624-630; *Adv. Mater.* **2022**, *34*, 2201844; *Sci. Adv.* **2021**, *7*, eabd9715). These works proved the enhanced luminance efficiency when insulating materials were embedded into the light-emitting semiconductors, and elucidated that the performance improvement was attributed to enhanced charge transporting characteristics of the emission layers by the chain aggregation. This is analogous to our results shown in **Supplementary Fig. 15**, exhibiting higher current density and luminance in the SI-OLEDs than pristine OLEDs. Consequently, enhanced charge transport of the SI-OLES induced by molecular aggregation is responsible for the improvement EL performance, indicating the novelty of our material design in that

micro-patternability and light-emitting performance can be obtained simultaneously.

To clarify the origin of the improved luminance efficiency of the SI-OLEDs, we added the following sentence and references on the page 9 and References, respectively, in the revised manuscript.

“This preserved or improved light-emitting performance of the SI-OLEDs is attributed to the molecular entanglement in the SI-OLES caused by the silicone network, leading to enhanced charge transport within the emission layers^{26,27,33-35}.”

[33] Zhang, Z. et al. High-brightness all-polymer stretchable LED with charge-trapping dilution. *Nature* **603**, 624-630 (2022)

[34] Liu, Y. et al. A Self-Assembled 3D penetrating nanonetwork for high-performance intrinsically stretchable polymer light-emitting diodes. *Adv. Mater.* **34**, 2201844 (2022)

[35] Kim, J.-H. & Park, J.-W. Intrinsically stretchable organic light-emitting diodes. *Sci. Adv.* **7**, eabd9715 (2021)

Q3) *When I checked the micropatterns of blue layers which do not have silicone-engineering in Fig. S17 – 19, the patterning results were not so different with red and green SI-OLES layers. It would be interesting to do supplementary experiments or characterization to emphasize the strong anisotropic etching in the manuscript.*

Response: We thank the comment from the reviewer. As the reviewer mentioned, although the micropatterns of the blue (B)-OLES may appear to be plausibly well-defined, when we took a look deeply, the pattern fidelity of the B-OLES (variation of pattern width and pattern edge roughness) was significantly different from that of the SI-OLES. As shown in **Figure R4**, the B-OLES micropatterns (2 $\mu\text{m} \times 6 \mu\text{m}$) exhibited much higher values of the width variation and edge roughness than the R- and G-SI-OLES. This indicates that a side-etching behavior was dominantly generated in the B-OLES, so that much smaller pattern than expected and severe side abrasion of the micropatterns were obtained. These results could be serious challenges in consecutive patterning processes and bring about low pixel uniformity. As the case of the R- and G-SI-OLES, the physico-chemical robustness of the B-OLES could be achieved by incorporating silicone network (**Figure R7**), but at the same time, incorporating silicone network accompanied considerable degradation in the luminescence characteristics for B-OLES (which even in its pristine form already exhibit inferior luminescence efficiency compared to pristine R- and G-OLES). Therefore, B-OLES but not the B-SI-OLES was used in this manuscript. We anticipate that development of high efficiency B-OLES would provide effective solution to realize the B-SI-OLES, thereby achieving highly reliable fine micropatterns of the B-SI-OLES.

To alleviate the reviewer’s concern, we added this figure as **Supplementary Figure 20** in the revised manuscript.

Figure R4 | Pattern fidelity of the B-OLES-based micropatterns. Evaluation of pattern width and pattern edge roughness of the B-OLES-based micropatterns, compared to the R- and G-SI-OLES.

Reply to Reviewer #3

The manuscript entitled "Silicone engineered anisotropic lithography for ultrahigh-density OLEDs" submitted by D. H. Kim and coworkers presented intriguing etching characteristics of silicone-incorporated organic light-emitting semiconductors (SI-OLES), enabling to be delicately pixelated by conventional photolithography and reactive ion etching. The approach that unique etching behavior of silicon is implanted on the OLED materials is particularly noteworthy, and this technique seems to significantly contribute improvement of pattern resolution in the field of the OLEDs. Also, the authors systematically described the non-volatile blocking layer that is induced during etching of the SI-OLES, as well as the developed high density RGB OLED pattern arrays and the light-emitting performance were quite impressive. In addition, the manuscript is well-written, and the figures are well-organized. I recommend this high quality manuscript for publication in Nature Communications after minor revision, and several points should be addressed before publication.

Response: We appreciate the positive comments from the reviewer. We revised the manuscript according to the reviewer's comments.

Q1) The authors mentioned that for analysis of non-volatile etch-blocking layer generated on the surface regime of the SI-OLES film, the lowest RF power of RIE was deliberately used. However, as the RF power is too low, the etching performance of RIE can be extremely reduced because ion bombardment effect is not efficiently developed. How can it confirm that the variation of surface modulus of the SI-OLES is the results from the etching reaction of RIE? Also, the authors should indicate the detailed RIE condition.

Response: We thank the reviewer’s comment and agree with your concern. To further confirm etching behavior under lowest RF power (10 W) condition of RIE system, we evaluated variations of film thickness of the R- and G-SI-OLES depending on RIE exposure times. As shown in **Figure R5**, all OLES and SI-OLES films showed slight reduction (about 7-9 nm) of film thickness under the RIE condition, indicating that etching behavior (in this case, chemical etching reaction) was apparently developed even under the lowest RF power we used (see **Supplementary Note 2**). This result implies that the ion bombardment effect of the RIE was minimized, so that the chemical RIE reaction (generation of non-volatile EBL and CH₂ scission reaction) was dominantly induced. In this regard, we believe that the variation of surface modulus of the SI-OLES was a result from the chemical etching reaction of the RIE.

Figure R5 | Film thickness of OLES and SI-OLES after exposure of RIE. a,b, Evaluation of thickness of the R- and G-OLES/SI-OLES films under the lowest radio frequency (RF) power condition of the RIE, as a function of etching time. The detailed RIE condition is as follows; Ar/O₂ (40 sccm/20 sccm) and 10 W RF power.

Q2) In the EL performance of OLEDs based on the OLES and SI-OLES, both R and G-SI-OLES showed higher current density than the pristine ones. This seems to importantly affect the preservation or improvement of luminous property of the SI-OLEDs, compared to pristine OLEDs. What is the origin of the enhanced electrical characteristics of the SI-OLES even though the insulating materials are embedded?

Response: We appreciate the reviewer’s comment. An initial mixing solution of the polymers and silicone

network precursors shows homogenous phase; however, as the ladder-like silicone network is constructing by sol-gel reaction, molecular micro-phase separation between the polymer chains and the assembling network occurs, so that thermodynamic molecular entanglement is spontaneously induced (**Supplementary Fig. 1**). As a result, 3-dimensional molecular entanglement between the silicone network and the polymer chains is achieved, indicating that the polymer chains are molecularly aggregated. The induced chain aggregation facilitates charge transporting capability, so that the electrical characteristics of the polymer semiconductor are not deteriorated even though insulating materials are introduced (*Adv. Mater.* **2019**, *31*, 1901400; *ACS Appl. Mater. Inter.* **2020**, *12*, 55107-55115). Recently, the molecular aggregation mechanism has been exploited in the field of polymer LEDs (*Nature* **2022**, *603*, 624-630; *Adv. Mater.* **2022**, *34*, 2201844; *Sci. Adv.* **2021**, *7*, eabd9715). These works presented the enhanced luminance efficiency when insulating materials were embedded into the light-emitting semiconductors, and elucidated that the performance improvement was attributed to enhanced charge transporting characteristics of the emission layers by the chain aggregation. This is analogous to our results shown in **Supplementary Fig. 15**, exhibiting higher current density and luminance in SI-OLEDs than pristine OLEDs. Consequently, enhanced charge transporting ability of SI-OLES induced by molecular aggregation is responsible for the improvement EL performance, indicating the novelty of our material design in that micro-patternability and light-emitting performance can be obtained simultaneously.

To clarify the origin of the improved luminance efficiency of the SI-OLEDs, we added the following sentence and references on the page 9 and References, respectively, in the revised manuscript.

“This preserved or improved light-emitting performance of the SI-OLEDs is attributed to the molecular entanglement in the SI-OLES caused by the silicone network, leading to enhanced charge transport within the emission layers^{26,27,33-35}. ”

[33] Zhang, Z. et al. High-brightness all-polymer stretchable LED with charge-trapping dilution. *Nature* **603**, 624-630 (2022)

[34] Liu, Y. et al. A Self-Assembled 3D penetrating nanonetwork for high-performance intrinsically stretchable polymer light-emitting diodes. *Adv. Mater.* **34**, 2201844 (2022)

[35] Kim, J.-H. & Park, J.-W. Intrinsically stretchable organic light-emitting diodes. *Sci. Adv.* **7**, eabd9715 (2021)

Q3) *Does the measured contact angle of the SI-OLES film correspond to either R-SI-OLES or G-SI-OLES? This is not clear in the manuscript.*

Response: We appreciate the comment from the reviewer. We reorganized the contact angle data of the R- and G-OLES/SI-OLES films with proper indications. To avoid misunderstanding, we replaced the original figure with this figure as **Supplementary Figure S14** in the revised manuscript.

Figure R6 | Comparison of contact angle between the OLES and the SI-OLES films. a, Contact angle of the OLES and the SI-OLES films with water droplet. **b,** A schematic of chemical structure of the SI-OLES film.

Q4) The author mentioned that the used Blue-OLES in this work was not incorporated with silicone network because the luminescence performance of the SI-B-OLES was too low, and this problem can be solved if high efficient Blue-OLES is exploited. However, even if high efficient blue materials are developed, it could be meaningless if the chemical robustness of bluish semiconducting materials is not demonstrated when silicone network is adapted. At least, the authors should provide additional data that even blue materials can possess high tolerance when silicone network is integrated.

Response: We appreciate the comment from the reviewer. As the reviewer mentioned, we carried out further experiments to investigate whether the adapted B-OLES in this work can obtain physico-chemical tolerance when the silicone network is embedded. As shown in **Figure R7**, the B-SI-OLES film showed almost 100 % film retention after solvent rinsing process. In conjunction with the chemical robustness, the etch rate of the B-SI-OLES was diminished by 21.7 %, compared to that of the B-OLES. These results indicate that the proposed SI-OLES material design can be also effective methodology to impart chemical and physical robustness into blue light-emitting polymer series. Consequently, we believe that our material design possesses a great potential to realize ultrahigh-density blue OLEDs with high light-emitting performance when high efficiency blue materials are developed.

To address the reviewer's concern, we added this figure as **Supplementary Figure 19** and the following sentence on the page 11 in the revised manuscript.

"Note that unlike the R- and G-SI-OLES, the B-OLES was not integrated with the silicone network in this study, even though the B- OLES was able to achieve high physico-chemical tolerance when the silicone network was embedded (Supplementary Fig. 19). This is because the B-OLES is based on a ternary

blending system (host-dopant-hole transporting materials) to enhance its luminescence efficiency; however, the incorporated silicone network seems to interfere energy transfer process within the B-OLES layer, so that the light-emitting performance significantly deteriorated.”

Figure R7 | Evaluation of chemical and physical robustness of the B-SI-OLES film. a, Film retention of B-SI-OLES after solvent rinsing process. **b,** Variation of film thickness of B-OLES and B-SI-OLES films were measured as a function of the RIE etching time based on Ar/O₂

Q5) Several typos should be corrected.

Response: We appreciate the reviewer for pointing out typing errors. We corrected the typos and reflected them in the revised manuscript.

REVIEWERS' COMMENTS

Reviewer #1 (Remarks to the Author):

The authors have addressed all my concerns.

The manuscript is now suitable for publication.

Reviewer #2 (Remarks to the Author):

I am satisfied with the revisions and have no further comments.

Reviewer #3 (Remarks to the Author):

The authors have addressed all comments raised by us. The quality of 2nd manuscript has been significantly improved.

- In the revised version, we can see more clear strategy compared to the previous works in the field of high density OLEDs.

- In conclusion, I think this work will serve a rational direction of real applications of OLEDs.

- The current version is ready for publishing.

Reviewers' Comments

Reviewer #1 (Remarks to the Author):

The authors have addressed all my concerns. The manuscript is now suitable for publication.

Response: We thank the reviewer for the positive assessment to our work.

Reviewer #2 (Remarks to the Author):

I am satisfied with the revisions and have no further comments.

Response: We appreciate the reviewer for the positive assessment to our work.

Reviewer #3 (Remarks to the Author):

The authors have addressed all comments raised by us. The quality of 2nd manuscript has been significantly improved.

- In the revised version, we can see more clear strategy compared to the previous works in the field of high density OLEDs.

- In conclusion, I think this work will serve a rational direction of real applications of OLEDs.

- The current version is ready for publishing.

Response: We thank the reviewer for the positive assessment to our work.

A brief summary of the main findings of this manuscript. (Maximum 250 characters, including spaces)

Description: Silicone-incorporated organic light-emitting semiconductors, inspired by etching nature of silicon, can achieve anisotropic lithography via reactive ion etching-coupled photolithography, resulting in development of ultrahigh-density RGB OLED arrays.